# Retrospective and Randomized Analysis of Influence and Correlation of Clinical and Molecular Prognostic Factors in a Mono-Operative Series of 122 Patients with Glioblastoma Treated with STR or GTR

**DOI:** 10.3390/brainsci10020091

**Published:** 2020-02-09

**Authors:** Maurizio Salvati, Placido Bruzzaniti, Michela Relucenti, Mariagrazia Nizzola, Pietro Familiari, Marco Giugliano, Anthony Kevin Scafa, Santi Galletta, Xiaobo Li, Rui Chen, Claudio Barbaranelli, Alessandro Frati, Antonio Santoro

**Affiliations:** 1Department of Neurological Sciences, Neurosurgey, “La Sapienza” University of Rome, 00161 Rome, Italy; maurizio.salvati@uniroma1.it (M.S.); grazia.nizzola@gmail.com (M.N.); pietro.familiari@uniroma1.it (P.F.); marco.giugliano@uniroma1.it (M.G.); ak.scafa@gmail.com (A.K.S.); antonio.santoro@uniroma1.it (A.S.); 2Department of Anatomical, Histological, Forensic Medicine and Orthopedic Science, “La Sapienza” University of Rome, 00161 Rome, Italy; Michela.relucenti@uniroma1.it; 3UOSD of Neurophysiopathology and DISMOV, AOU G Martino, Department of Clinical and Experimental Medicine, University of Messina, 98122 Messina, Italy; santi.galletta@polime.it; 4Key Laboratory of Environmental Medicine Engineering, Ministry of Education, School of Public Health, Southeast University, Dingjiaqiao 87, Nanjing 210009, China; xiaobo.li@njit.edu.cn (X.L.); chenrui@njit.edu.cn (R.C.); 5Department of Psychology, Faculty of Medicine and Psychology “La Sapienza” University of Rome, 00189 Rome, Italy; claudio.barbaranelli@uniroma1.it; 6Department of Neurosurgery, IRCCS Neuromed Pozzilli IS, 86077 Isernia, Italy; alessandro.frati@uniroma1.it

**Keywords:** Glioblastoma multiforme, GTR, STR, KPS, IDH1, MGMT, ATRX, EGFR, TP53, Ki67, Neurosurgery, Oncology, Epilepsy

## Abstract

Glioblastoma is a solid, infiltrating, and the most frequent highly malignant primary brain tumor. Our aim was to find the correlation between sex, age, preoperative Karnofsky performance status (KPS), presenting with seizures, and extent of resection (EOR) with overall survival (OS), progression-free survival (PFS), and postoperative KPS, along with the prognostic value of IDH1, MGMT, ATRX, EGFR, and TP53 genes mutations and of Ki67 through the analysis of a single-operator series in order to avoid the biases of a multi-operator series, such as the lack of homogeneity in surgical and adjuvant nonsurgical treatments. A randomized retrospective analysis of 122 patients treated by a single first operator at Sapienza University of Rome was carried out. After surgery, patients followed standard Stupp protocol treatment. Exclusion criteria were: (1) patients with primary brainstem and spinal cord gliomas and (2) patients who underwent partial resections (resection < 90%) or a biopsy exclusively for diagnostic purposes. Statistical analysis with a simultaneous regression model was carried out through the use of SPSS 25^®^ (IBM). Results showed statistically significant survival increase in four groups: (1) patients treated with gross total resection (GTR) (*p* < 0.030); (2) patients with mutation of IDH1 (*p* < 0.0161); (3) patients with methylated MGMT promoter (*p* < 0.005); (4) patients without EGFR amplification or EGFRvIII mutation (*p* < 0.035). Higher but not statistically significant survival rates were also observed in: patients <75 years, patients presenting with seizures at diagnosis, patients affected by lesions in noneloquent areas, as well as in patients with ATRX gene mutation and Ki-67 < 10%.

## 1. Introduction

Glioblastoma (GBM) is a solid, infiltrating, highly malignant tumor, and a grade IV glioma according to 2016 World Health Organization (WHO) classification [1]. It is believed that GBM is derived from a small population of cancer cells known as glioma stem cells (GSCs) and that these derive from the uncontrolled proliferation of neuronal stem cells (NSCs) residing in restricted germinal areas: ventricular subependymal zone of the temporal horn of the lateral ventricle (SVZ), the subgranular zone of the dentate gyrus of the hippocampus (SGZ), and of the white subcortical substance [2]. It is the most frequent malignant primary brain tumor (16% of all primitives of the CNS and 54% of glial tumors) [3,4]. The average survival from diagnosis is less than 15 months, with survival rates at between 26%–33% at two years and 3%–10% at five years [5,6]. The development of GBM involves molecular pathways, which are different in primary and secondary lesions [7,8]. Standard treatment provides for maximum surgical resection followed by conformational radiotherapy (~60 Gy/30 fractions) for up to six weeks, concomitant with temozolomide (75 mg/m^2^/day) and then maintenance therapy with standard temozolomide schedule (150–200 mg/m^2^ × 5 days, every 28, for 12 cycles) [6]. We used Levetiracetam as a prophylactic treatment for seizures instead of valproate as described by other authors [9]. The objective of this retrospective study was to clarify the influence and correlation of prognostic clinical and molecular factors on survival and quality of life of GBM patients in a mono-operative series, thus avoiding biases related to a multioperator casuistry with the lack of homogeneity of adjuvant treatments. The study also specifically investigated the independence or correlation of the variables examined by means of a multivariate analysis performed with a simultaneous regression model.

## 2. Results

The average overall survival (OS) and progression-free survival (PFS) of the population studied were 23.70 (STD 18.64) and 9.93 (STD 11.86) months, respectively. The average preoperative Karnofsky performance status (KPS) was 83.48 (STD 15.83), while the average postoperative KPS was 80.11 (STD 16.51).

In total, 63.64% of the sample was male with an average age of 54.28 years (STD 13.86), average OS of 23.46 months (STD 17.71), PFS 8.70 months (STD 12.10), and postoperative KPS of 79.10 (STD 19.43), while 36.36% was female with an average age of 60.31 years (STD 12.18), average OS of 24.12 (STD 20.77), PFS 12.06 (STD 11.06), and postoperative KPS of 81.87 (STD 9.81). We found no statistically significant differences between males and females in terms of outcome.

Patients under 50 years of age had an OS of 22.76 months (STD 13.59), a PFS of 9.61 months (STD 9.83), a preoperative KPS of 83.07 (STD 23.93), and a postoperative KPS of 82.06 (STD 17.02); patients aged 51 to 75 years had an OS of 26.19 months (STD 21.74), a PFS of 11.11 (STD 13.71), a preoperative KPS of 84.4 (STD 11.48), and an average postoperative KPS of 78.26 (STD 17.6); patients over 75 years of age had an OS of 13.80 months (STD 6.41), a PFS of 4.40 months (STD 1.81), a preoperative KPS of 78 (STD 8.36), and a postoperative KPS of 84 (STD 5.47), Figure 1.

Patients with preoperative KPS greater than or equal to 80 points had an OS of 26.44 months (STD 20.31) and a PFS of 11.52 months (STD 13.04); patients with preoperative KPS between 50 and 80 points had an OS of 13.25 months (STD 4.06) and a PFS of 4.62 months (STD 2.92); patients with preoperative KPS less than 50 points (only 1.32% of our population) had an average OS of 15.66 months (STD 7.63) and a PFS of 4 months (STD 1.81).

Half of our population started with epileptic seizures. The mean age of patients who started with seizures was 52.40 years (STD 12.58), OS 26.36 months (STD 22.05) vs. 21.04 (STD 14.52) with *p*-value = 0.350 and a mean PFS of 8.86 months (STD 11.56) vs. 11 (STD 12.33) in the control group. Preoperative KPS in patients with seizures was 81.13 (STD 19.87) vs. 85.95 (STD 9.95) of the controls with a *p*-value = 0.324. Postoperative KPS in patients with seizures was 80.22 (STD 18.15) vs. 80 (STD 15.11) of the controls and a *p*-value = 0.964, Figure 2.

Patients undergoing gross total resection (GTR) had an average OS of 27.61 months (STD 20.38) and a PFS of 11.87 (STD 13.52). The control group underwent a subtotal resection (STR) with an OS of 14.38 months (STD 8.57) and a PFS of 5.30 months (STD 3.77), Figure 3. The Student *t*-test showed a statistically significant difference between the OS of the two groups (27.61 months vs. 14.38 months, *p*-value = 0.030), while the difference in PFS was remarkable but not significant (11.87 months vs. 5.31 months, *p*-value = 0.091).

The difference between postoperative KPS in the GTR and STR groups was not statistically significant. However, the distribution of the sample showed that the STR group presented medium–high postoperative KPSs (between 60 and 100), while in the GTR group there were patients with low postoperative KPSs, as shown in Figure 4.

Patients with wild type IDH1 gliomas had an OS of 22.63 months (STD 17.67), while for the mutated IDH1 group it was 38.33 (STD 29.73), *p*-value = 0.016. PFS in the wild-type group was 10.07 months (STD 12.25) vs. 8 months in the mutated group (STD 4.00), *p*-value = 0.774. Preoperative KPS for the wild type group was 83.50 (STD 16.06), while for the mutated IDH1 group it was 83.33 (STD 15.27), *p* = 0.980. Postoperative KPS for the wild-type group was 70 (STD 36.05), while for the mutated IDH1 group it was 82.31 (STD 12.45), *p*-value = 0.105.

Patients with MGMT methylation (Figure 5) had an OS of 31.95 months (STD 5.19), while for patients without its methylation it was 16.83 months (STD 2.01), *p*-value = 0.005. The mean preoperative KPS in patients with promoter methylation was 87.75 (STD 11.75), whereas in patients without methylation it was 79.78 (STD 18.12), *p*-value = 0.100. The mean postoperative KPS in patients in the methylation group was 77.50 (STD 19.15), whereas in the methylation-free group it was 82.29 (DS 13.98) *p*-value = 0.343.

Patients with ATRX loss (Figure 6) had an OS of 30.75 months (STD 29.62), while patients without mutation had an OS of 22.13 months (STD 15.42) *p*-value = 0.241. The PFS of the group with the ATRX loss was 14.12 months (STD 17.51), while in the control group it was 9 months (STD 10.33), *p*-value = 0.274. The group of patients with ATRX loss had a preoperative KPS of 82.50 (STD 11.64) vs. a KPS of 83.71 (STD 16.77) in the normal type, *p* = 0.847. The ATRX loss group had a postoperative KPS of 76.25 (STD 16.85) vs. a KPS of 80.97 (STD 16.55) of the normal type ATRX group, *p*-value = 0.470.

Patients with EGFR amplification/EGFRvIII mutation (Figure 7) had a mean OS of 19.83 months (STD 16.31) vs. 32 months (STD 21.15) of the control group with *p*-value = 0.035. The mean PFS in the EGFR amplification/EGFRvIII mutation group was 7.4 months (STD 7.84), whereas in the control group it was 15.75 (STD 16.79) *p*-value = 0.055. Preoperative KPS was 81.50 (STD 17.27) in the EGFR amplification/EGFRvIII mutation group vs. 88.07 (STD 11.08), *p*-value = 0.214, while average postoperative KPS was 80.51 (STD 12.94) vs. 77.14 (STD 22.97), *p*-value = 0.421.

In total, 82.67% of our population presented with TP53 mutation, with an OS of 21.13 months (STD 13.42), a PFS 8.96 months (STD 11.05), and a postoperative KPS 80.97 (STD 16.77). In the 17.33% without TP53 mutation, we found an OS of 27 (STD 19.33), a PFS of 11.21 (STD 12.45), and a KPS of 78.35 (STD 19.35). No statistically significant differences were found between the two groups (OS *p*-value = 0.156).

The sample with Ki-67 ≤ 10% had an average OS of 31.69 months (STD 19.17), an average PFS of 13.15 months (STD 13.52), preoperative KPS of 90 (STD 11.28), and postoperative KPS of 76.92 (STD 22.13); in patients with 10% < Ki-67 ≤ 20%, we found an average OS of 28.46 months (STD 24.68), an average PFS of 10.93 months (STD 13.02), preoperative KPS 79 (STD 21.23), and postoperative KPS 75.66 (STD 24.70); the population with a Ki-67 > 20% had an average OS of 16.33 months (STD 11.89), a PFS 7.26 months (STD 8.95), preoperative KPS 84.33 (STD 12.08), and postoperative KPS 81.33 (STD 9.90). Comparing the group with Ki-67 ≤ 10% and the group with Ki-67 > 20%, we observed an OS of 31.69 vs. 16.33 months respectively with *p*-value = 0.021, Figure 8.

Multivariate analysis showed that more than 50% of the OS of our population depended on the variables examined (*R*^2^ = 0.496, F(9,34) = 3.723, *p* = 0.002). The number of months between the first procedure and the recurrence of disease was significantly associated with OS (B = 0.313, t = 3.213, *p* = 0.003). The percentage of Ki-67 showed an association with OS tending to statistical significance (B = −0.025, t = −1.816, *p* = 0.078).

The independent variables examined as a whole also statistically significantly correlated for about 50% with PFS (*R*^2^ = 0.496, F(9,32) = 3.493, *p* = 0.004).

## 3. Discussion

We examined, in a consecutive single-operator series of 122 GBM patients surgically treated from 2013 to 2017 at Sapienza University of Rome, the correlation between sex, age, preoperative KPS, presenting with seizures, and extent of resection (EOR) with OS, PFS, and postoperative KPS, along with the prognostic value of mutations of IDH1, MGMT, ATRX, EGFR, and TP53 genes and of Ki67.

Our study, therefore, carried out a systematic and complete analysis of the main prognostic factors clinical and molecular of GBM on a very homogeneous patient series treated by a single first operator in a single institution. Simultaneous and multivariate analysis allowed us to investigate the correlation with the prognosis and quality of life.

Our results showed that sex did not affect prognosis. Sex influences survival only when combined with the methylation state of the MGMT promoter: women with a methylated phenotype have a higher OS than men with the same phenotype [10].

OS and PFS were instead significantly higher in the group of patients younger than 75 years of age (age ≥ 75 years was an independent negative prognostic factor). Also, the group of patients with preoperative high KPS (KPS ≥ 80) showed a significantly better prognosis.

Presenting with seizures was found to be remarkable, as well. OS and PFS were, in fact, higher in the group with seizures than in the group without seizures [11]. This finding might be related to the higher chance of an early diagnosis being more frequent in cortical lesions and in mutated IDH1 gliomas. The mutation of IDH1 (see below) leads, as is well known, to the formation of 2-HG (2-hydroxyglutarate). 2-HG has a molecular structure similar to glutamate and is able to bind and activate *N*-methyl-*D*-aspartate receptors (NMDA), thus resulting in a possible reduction in seizure threshold (this mutation is associated with about 70%–88% of low-grade gliomas with these being more epileptic than high-grade ones). Antiepileptic therapy could also have a sensitizing role in chemotherapy treatment, as pointed out by Vecth et al. [11]. Recently, it has been found that by combining valproic acid (VPA) with TMZ, survival rates improve in adults with GBM as well as children with brain tumors other than GBM. This could possibly be explained by the chemotherapy-sensitizing properties of VPA, including the inhibition of histone deacetylase, leading to improved survival [11]. The numerous side effects, drug interactions, and the consequent poor handling of this drug compared to others such as Levetiracetam (LEV), used for our patients, must be, however, taken into account. Moreover, LEV can provide a survival benefit in patients with GBM who receive TMZ [12]. Other authors suggested that the clinical therapeutic efficacy of TMZ in GBM might be potentiated through the combination with LEV and the enhancement of apoptotic pathways [13]. Nonetheless, a combined analysis of survival association with antiepileptic drug use at the beginning of chemo-radiotherapy and TMZ, performed in a pooled patient cohort (*n* = 1869) of four contemporary randomized clinical trials in newly diagnosed GBM, showed no outcome improvement for either LEV or VPA use [14]. It is therefore necessary to carry out a prospective study to clarify whether the use of LEV or VPA associated with TMZ is able to determine a real improvement in the outcome, and which of the two drugs is more effective. Our study confirmed that the increase in OS in patients treated with LEV, though present, had no statistical significance and it was rather due to the characteristics of the tumors presenting with seizures: early diagnosis and cortical location, which makes these lesions more accessible.

As far as surgery is concerned, the GTR group had significantly higher OS and PFS than the STR group, as widely reported in current literature [15]. Patients with lesions in eloquent areas were treated with STR in order to avoid neurological deficits resulting in a postoperative KPS and quality of life worsening (Figure 8). In our series, none of the patients in the STR group exceeded 36 months of OS, while in the group of patients treated with GTR there were survival rates of up to 85 months (none of the patients were treated with GTR in reintervention after being treated with STR, as reported by Block O et al. [16]). When STR was performed, residual disease volume seemed to be the most important factor influencing OS and PFS [17], being—in our series—an independent prognostic factor. We did not find any difference in terms of survival in patients with residual tumor when the extent of resection exceeded 90%. Our results showed that survival was markedly greater not only in patients treated with GTR as a first intervention, but also in patients who were treated with GTR in reintervention. GTR is hence an independent prognostic factor for survival, even in reintervention cases and even if associated with lower postoperative KPS values in comparison to STR.

Multivariate analysis of prognostic factors showed that the number of months between tumor removal and disease recurrence was the independent variable most specifically related to OS (B = 0.313, t = 3.213, *p* = 0.003) and was also related to postoperative KPS. This parameter summarizes the validity of the treatment and can be influenced by the presence of tumor residue [18] and by resistance to treatment with temozolomide [19]. This parameter may, therefore, be fundamental in the selection of recurrences amenable to a new surgical treatment and in the choice of a new chemotherapy line.

We also evaluated prognostic value of IDH1, MGMT, ATRX, EGFR, and TP53 gene mutations, and of Ki67.

Mutation of IDH1 gene was found in less than 10% of the sample and was associated, in line with current literature, with a significant increase in OS and PFS (without statistically significantly affecting postoperative KPS).

A recent metanalysis pointed out the scarcity of evidence in terms of a direct relationship between methylated MGMT promoter and PFS [20]. The results of our study showed that MGMT promoter methylation was an independent positive prognostic factor for both OS and PFS, but—as for IDH1—not a predictive factor for postoperative KPS. This was, however, the molecular marker that correlated the most with survival.

A similar phenotype was also induced by ATRX loss and low-ATRX mRNA expression which, in our study, were associated with an increase in survival, though without statistical significance. This finding is in agreement with the Jiao report [21].

In their study, Ramamoorthy et al. proved that in the absence of ATRX, the histone variant macroH2A1.1 binds to the polymerase tankyrase 1, preventing it from localizing to telomeres and resolving cohesion, thus promoting recombination between sister telomeres. Forced resolution of this event induces genomic instability, thereby impeding cell growth [22].

Liu et al. [23] highlighted the absence of ATRX within secondary glioblastomas, and more particularly in younger patients, whereas Cai et al. [24] observed a higher rate of lower ATRX expression in primary GBM and grade III gliomas than in grade II gliomas, and suggested this as a malignancy marker.

Our study showed that ATRX loss and low-ATRX mRNA expression play an important role, not only in the survival of patients affected by LGG, but also in case of HGG. These findings may be a potential therapeutic target for high grade glial tumors, hence the need for further investigations.

EGFR amplification/EGFRvIII mutation was—in our population—a negative independent prognostic factor in terms of OS, presenting a trend towards statistical significance for PFS too. Nonetheless, no improvement in survival was found with the EGFRvIII Rindopepimut^®^ vaccine [25]. The addition to the standard therapy of Nimotuzumab^®^, a humanized therapeutic monoclonal antibody against EGFR, yielded, on the other hand, positive results only in a post hoc analysis where it revealed an improvement in survival in patients with residual tumor and nonmethylated MGMT (PFS 6.2 vs. 4 months; OS 19 vs. 13.8 months). This could be due to receptor interference and associations with multiple transduction pathways and proteins of invasion and angiogenesis regulation and the development of resistance mechanisms. Therefore, new therapies are focused on a combination of targeted gene therapy against EGFR and EGFRvIII and transduction pathways and proteins related to this pathway [26].

EGFR amplification/EGFRvIII mutation, like the previous markers, did not appear to influence postoperative KPS.

Our study, with the advantage of investigating a large population through the elimination of the main confounding factor represented by multioperator treatment, showed that—in spite of the negative results of the clinical trials over Rindopepimut^®^ vaccine and Nimotuzumab^®^—there is solid clinical evidence of the role of EGFR amplification/EGFRvIII mutation in OS and PFS. Therefore, new clinical trials trying to block the EGFR signal transduction pathway at different levels in order to reduce resistance to therapy may be fundamental, and increased survival with anti-EGFR drugs in patients with nonmethylated MGMT should be particularly investigated. Recently, new EGFR-targeted therapies have been proposed, e.g., depatuxizumab mafodotin, which completed Phase I in a study with recurrent GBM patients with EGFR amplification and entered Phase III in the RTOG 3508 trial [27] as an adjunct therapy to standard therapy. In addition, a phase I study with T cells activated with a chimerical antigen against EGFRvIII shows good treatment tolerance and encouraging results [27,28].

TP53 is one of the most commonly deregulated genes in cancer. Deregulated p53 pathway components have been implicated in GBM cell invasion, migration, proliferation, evasion of apoptosis, and cancer cell stemness. Recent studies show that mutant TP53 is also strongly associated with a poor prognosis in terms of overall survival and with a decrease in chemosensitivity of GBM to TMZ by increasing MGMT expression [29]. In our study, TP53 mutation was associated with shorter OS and PFS, even though without statistical significance.

Ki-67 is a nonhistone nuclear protein, which is expressed throughout all active cell cycle phases, but not in the resting cell phase, G0. We evaluated the relationship between Ki-67 index and the outcome of patients. The analysis revealed that OS and PFS were inversely related with Ki-67 index. We divided our patients into three groups based on this parameter (Figure 6): <10%, between 10% and 20% and >20%. We found a statistically significant difference in terms of survival in the group with Ki-67 < 10% compared to the group with Ki-67 > 20% (*p*-value = 0.021).

In a recent report, Alkhaibary et al. [30] reported similar results, but in a series of 44 multioperator patients. Our investigation highlighted the strong role of Ki-67 as an unfavorable prognostic factor in GBM.

Postoperative KPS was not related to any of the independent variables examined. The only variable tending to statistical significance was, as previously reported, the number of months between tumor removal and recurrence, which was also the variable related the most to OS in the multivariate analysis [31].

## 4. Materials and Methods

A randomized retrospective analysis was performed on 122 patients with histological diagnosis of supratentorial GBM, treated from January 2013 to December 2017 at the Department of Neurosurgery of Sapienza, University of Rome. Preoperative study included an objective neurological examination with evaluation of KPS score and a radiological study performed with MRI 3T after the administration of gadolinium with the integration of DWI, PWI sequences, and spectroscopy. In the case of patients with a lesion in an eloquent area, a functional MRI was performed. The extension of resection was determined by comparing contrast-enhanced MRI images acquired within 24 h after surgical treatment with the preoperative ones, and calculated with the ABC/2 method. All the patients received an antiepileptic prophylaxis (Levetiracetam 1000 mg at the induction and then 500 mg as a maintenance therapy bid for 6 months).

### 4.1. Characteristics of the Patients

Characteristics of the population studied are summarized in Table 1.

Our population consisted of 76 males and 46 females. The age ranged between 31 and 82 years with an average of 56.30 years. In total, 77.27% of our population had a preoperative KPS greater than or equal to 80 points, this group had an average age of 55.61 years; 18.18% of the population had a preoperative KPS less than 80 points and greater than or equal to 50 with an average age of 59.37 years; only 1.32% of patients had a preoperative KPS less than 50 points with an average age of 64.66 years. Half of our population presented with seizures, treated with antiepileptic therapy. The average age of the group with epileptic seizures was 52.40 years.

### 4.2. Characteristics of the Tumors

In total, 59.09% of the tumors were located in the left hemisphere with the following sites in order of frequency: frontal lobe (45.45%), temporal lobe (25.03%), parieto-occipital (15.90%), and parieto-insular-occipital (2.27%). A total of 34.09% of the tumors involved eloquent areas.

The molecular characteristics are shown in Table 2.

The status of IDH and ATRX and the expression of EGFR, TP53, and Ki-67 were evaluated with an immunohistochemical technique, whereas the analysis of DNA methylation was performed by PCR or Southern blotting. In total, 91.18% of the sample had a wild-type IDH1 status and 45.44% of patients presented methylation of MGMT promoter. In 36% of patients there was ATRX loss and 59.08% of the cases showed EGFR overexpression. TP53 was not expressed in 36.36% of the cases, overexpressed in 27.20%, and focally expressed in 13.60%.

### 4.3. Treatment Characteristics

Surgery was performed by the same first operator with the aid of the following: neuronavigation system, intraoperative ultrasound, ultrasound aspirator (CUSA-CAVITRON^®^), thulium laser, and intraoperative neurophysiological monitoring. Of the procedures, 19.40% were conducted in awake surgery so as to monitor, real-time, the functions of the patient during surgery in eloquent areas. GTR (Figure 9) was performed in 85.95% of cases, while STR (Figure 10) in the residual 14.05%. Partial resections (resection < 90%) and biopsies were not included. Our follow-up consisted of radiological evaluation through brain MRI with gadolinium 20 days after surgery and subsequent clinical reevaluation. Cases of recurrence were also treated in our department: 66.54% of patients underwent a second procedure and 24.95% underwent three.

### 4.4. Statistical Analysis

Univariate analyses for sex, age, KPS, seizures at diagnosis, GTR/STR, IDH1, MGMT, ATRX, EGFR, TP53, and Ki-67 compared to OS were conducted with Kaplan–Meier curves and compared with log-rank tests, χ(chi)^2^, or *t*-student tests depending on the variables taken into account.

Multivariate statistical analysis was then carried out with bootstrap regression, conducting different regression analyses on the data. The independent variables used were sex, age, preoperative KPS, EOR (GTR or STR), months between 1st and 2nd surgery, methylation of MGMT gene promoter, EGFR over-expression, TP53 mutation, and percentage of Ki-67. The dependent variables were OS, PFS, and postoperative KPS. The so-called “enter” method was used in all regressions. It corresponds to a simultaneous regression model in which all the independent variables are simultaneously introduced into the regression equation [17]. The latter was conducted using the IBM^®^ SPSS 25 statistics software.

## 5. Conclusions

On the basis of the results presented, GTR can be considered the only treatment able to allow more than 36 months survival in patients with GBM, though potentially related with postoperative onset of neurological deficits is a worsening in the quality of life. Apart from surgical considerations, molecular biology of GBM plays a fundamental role. IDH1 mutation is a well-known positive prognostic factor for OS and PFS. It is, however, present in a small percentage of patients. MGMT gene promoter methylation is rather more important, being the independent molecular positive prognostic factor related the most with OS and PFS. ATRX loss and low-ATRX mRNA expression influence OS, though having a less remarkable impact on PFS. EGFR amplification/EGFRvIII mutation is a negative prognostic factor for both OS and PFS but the clinical trials performed so far have not been significantly effective. Ki-67 is inversely related to survival: patients with Ki-67 < 10% demonstrate a better outcome.

Multivariate data analysis showed that more than 50% of OS and PFS in the GBM population depends on the variables examined, while there is less correlation between these variables and postoperative KPS. The interval, in months, between tumor removal and recurrence of disease summarizes the efficacy of treatment and indicates the possibility for further intervention, as in our study this was the clinical parameter that more clearly correlates with OS and postoperative KPS.

Further prospective studies and clinical trials are necessary to evaluate, especially in patients without methylation of the MGMT promoter, the efficacy of therapies against EGFR mutations and their combination to stop this pathway at different levels. Treatment of glioblastoma should be therefore targeted according to molecular features.

## Figures and Tables

**Figure 1 brainsci-10-00091-f001:**
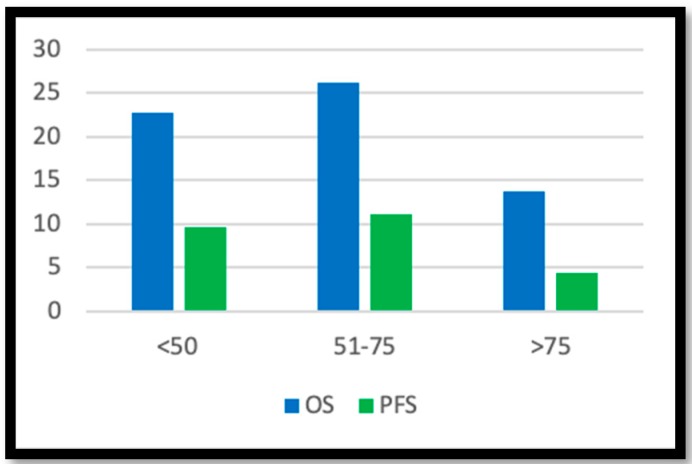
Overall survival (OS) and progression-free survival (PFS) in age-related groups.

**Figure 2 brainsci-10-00091-f002:**
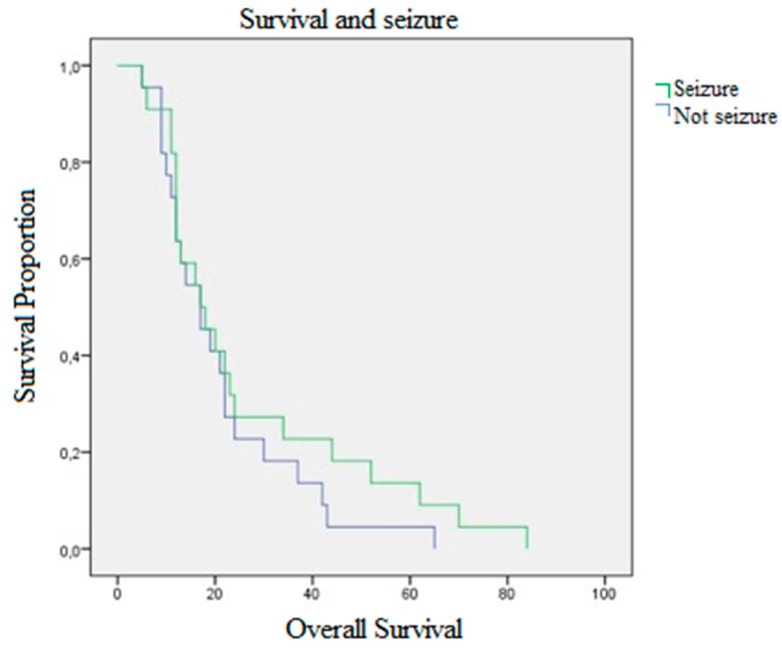
OS and seizures. In green, patients who with seizures at diagnosis, and in purple, patients who had no seizures.

**Figure 3 brainsci-10-00091-f003:**
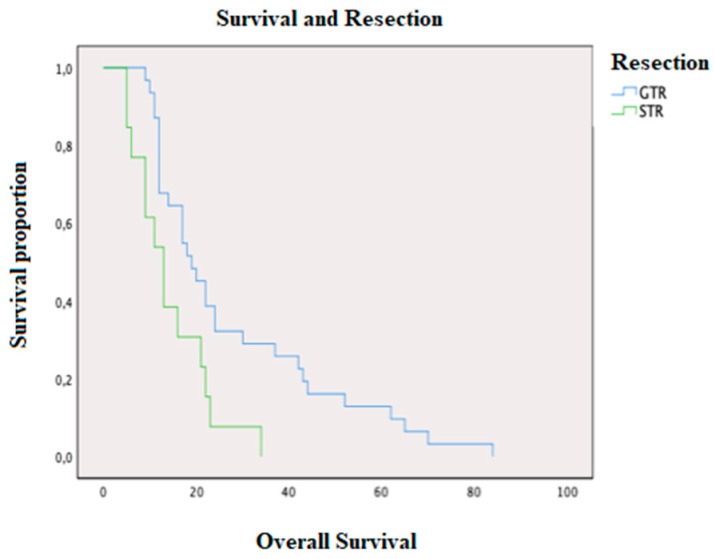
OS in patients treated with gross total resection (GTR; in blue) and subtotal resection (STR; in green).

**Figure 4 brainsci-10-00091-f004:**
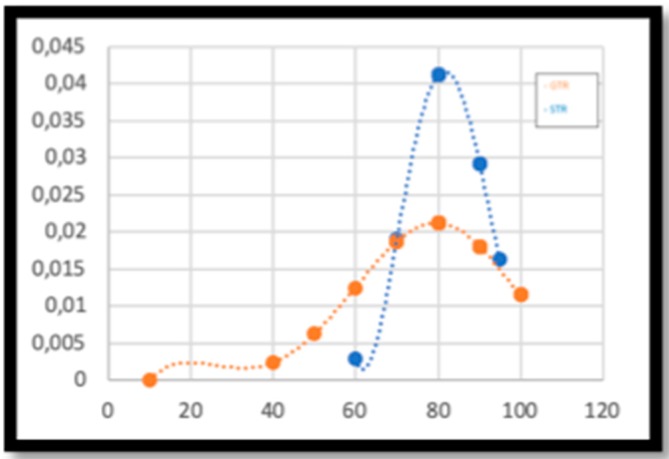
Distribution, in relation to Karnofsky performance status (KPS), of patients treated with STR (in blue) and patients treated with GTR (in orange).

**Figure 5 brainsci-10-00091-f005:**
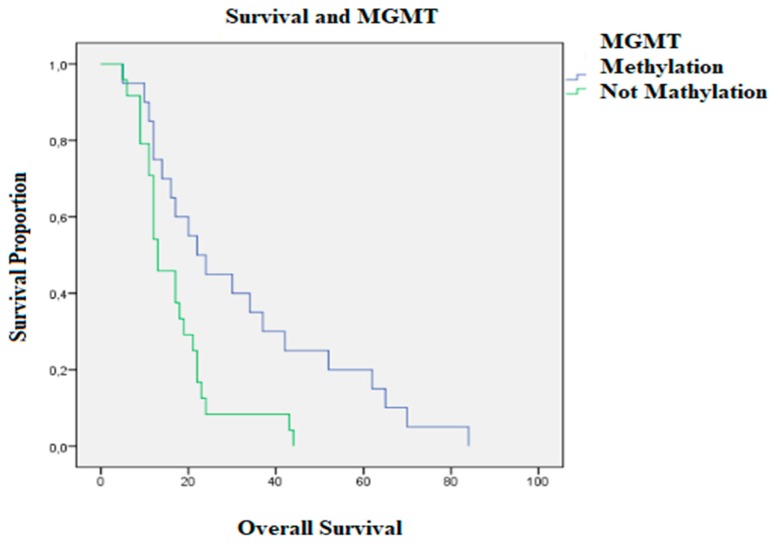
Survival of patients with methylation of the MGMT promoter (in purple) and patients without methylation (in green).

**Figure 6 brainsci-10-00091-f006:**
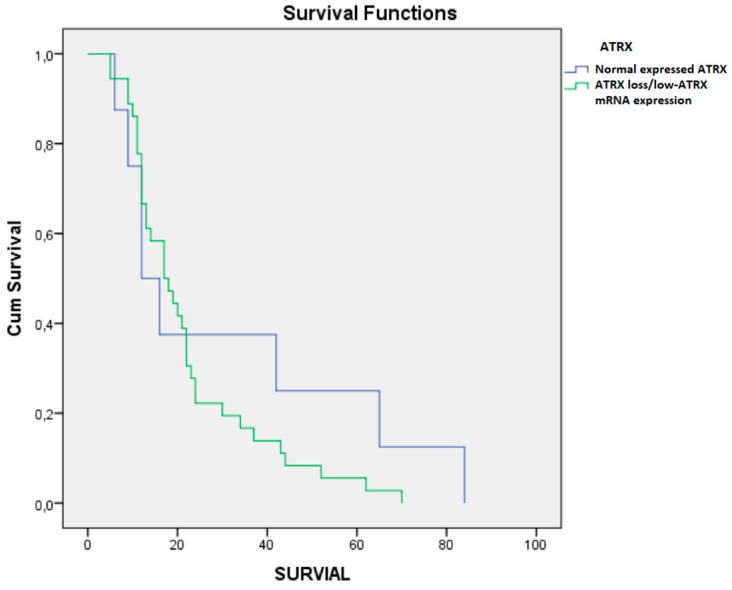
Survival of patients with ATRX loss (in blue) and patients without ATRX loss (in red).

**Figure 7 brainsci-10-00091-f007:**
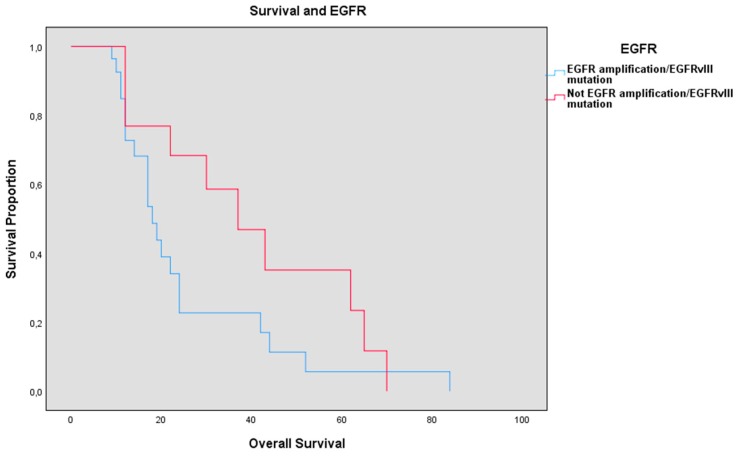
Survival of patients with EGFR amplification/EGFRvIII mutation (in blue) and patients without EGFR amplification/EGFRvIII mutation (in red).

**Figure 8 brainsci-10-00091-f008:**
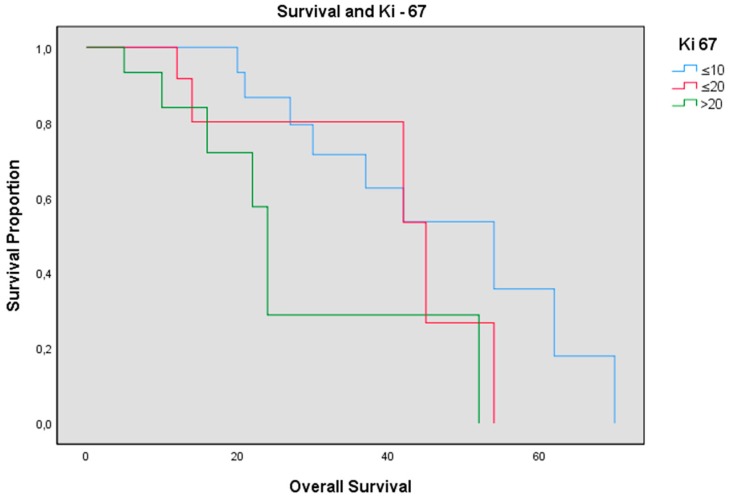
OS and Ki-67. In blue are patients with Ki-67 ≤ 10%, in red are patients with 10% < Ki67 ≤ 20%, and in green are patients with Ki-67 > 20%.

**Figure 9 brainsci-10-00091-f009:**
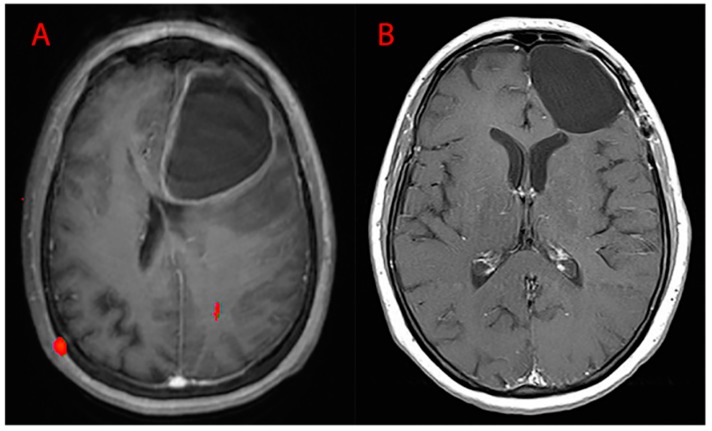
MRI T1WI axial sections. (**A**) Preoperative left frontal lobe GBM and (**B**) postoperative images (absence of residual disease, GTR).

**Figure 10 brainsci-10-00091-f010:**
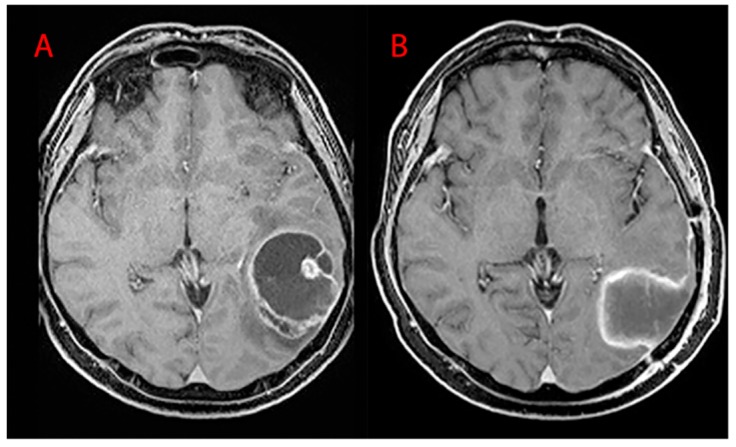
MRI T1WI axial sections. (**A**) Preoperative left parieto-occipital GBM and (**B**) postoperative images (EOR = 90%–100%, STR).

**Table 1 brainsci-10-00091-t001:** Characteristics of patients.

Variable		%
Sex		
	Male	59.01
	Female	40.99
Age (years)		
	≤50	29.54
	51–75	59.10
	>75	11.36
Preop-KPS		
	≤50	1.32
	50–79	18.18
	>80	77.27

**Table 2 brainsci-10-00091-t002:** Molecular characteristics.

Variable		%
Wild type IDH 1		91.18
Mutated IDH 1		8.82
Methylated MGMT		45.44
Nonmethylated MGMT		54.56
ATRX loss		36.01
Overexpressed EGFR		59.08
TP53 loss		36.36
Hyperexpressed TP53		27.20
Focally expressed TP53		13.60
Ki-67		
	0–10	2.54
	10–20	34.10
	>20	36.36

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
