# Peer review of "Retrospective and Randomized Analysis of Influence and Correlation of Clinical and Molecular Prognostic Factors in a Mono-Operative Series of 122 Patients with Glioblastoma Treated with STR or GTR"

_brainsci, 2020, doi:10.3390/brainsci10020091_

Round 1

Reviewer 1 Report

While this report is well written and logically conclusive and includes a substantial number of patients (N=122) from a single institution, there is unfortunately no scientific knowledge at all gained from this analysis. In this retrospective study  you re-analyze commonly known molecular factors and predictors of progression and show results that are consistent with previous results from randomized prospective trials and larger cohorts.

Author Response

Dear reviewer,

We understood your point of view and your considerations.

In our revision we remarked the importance of having analyzed a single-first operator series because this kind of approach eliminates the main bias in the evaluation of prognostic factors represented by the lack of homogeneity in the treatment of patients affected by GBM. We also tried to highlight that the role of many of these prognostic factors is still debated.  

Our series is perhaps one of the largest on this topic analyzing simultaneously both clinical and molecular prognostic factors and their correlation.

Best regards,

the authors

Reviewer 2 Report

Interesting work by the authors but the clinical significance and the impact of this paper is not expected to be as important. 

Author Response

Dear reviewer,

We understood your point of view and your considerations.

In our revision we remarked the importance of having analyzed a single-first operator series because this kind of approach eliminates the main bias in the evaluation of prognostic factors represented by the lack of homogeneity in the treatment of patients affected by GBM. We also tried to highlight that the role of many of these prognostic factors is still debated.  

The impact of our study is very important. Our series is perhaps one of the largest on this topic analyzing simultaneously both clinical and molecular prognostic factors and their correlation. The role of EGFR amplification/EGFRvIII mutation - though questioned by different clinical trials - is, for instance, clarified by our research, indicating the need for further investigations for the possibility to block EGFR pathway at various levels so as to design new targeted therapies.

Best regards,

the authors

Reviewer 3 Report

Dear authors,

thank you for submitting your manuscript "Retrospective and randomized analysis of influence and correlation of molecular and clinical prognostic factors in a mono-operative series of 122 patients with glioblastoma treated with STR or GTR."

In the introduction you pointed out that the objective of your study is a correlation to Quality of life. I miss this Point in your discussion. Please include it here.

Further you say, you treat your patients with Levetiracectam instead of Valproate. Could you also describe the advantage of Levetiracectam over Valproate.

Author Response

Dear reviewer,

We understood your point of view and your suggestions and we promptly took them into account in our revision.

Best regards,

the authors